

# Simulating the Tone River Eastward Diversion Project in Japan Carried Out Four Centuries Ago

Joško Trošelj[1,2*] and Naota Hanasaki[1]

5   [1] Center for Climate Change Adaptation, National Institute for Environmental Studies (NIES), Japan

[2] Laboratory for Informatics and Environmental Modelling, Division for Marine and Environmental Research, Institute Ruđer Bošković (IRB), Croatia

*Correspondence to:* Joško Trošelj (troselj.josko@nies.go.jp)

*** Secondary contact:* Naota Hanasaki (hanasaki@nies.go.jp)



**Abstract**

The Tone River is the largest river in Japan, flowing from the Kanto Plain westward to the Pacific Ocean. The river originally flowed southward, entering Tokyo Bay, but the Tone River Eastward Diversion Project in the 17th century and many later projects changed the flow route to that of today. The gradual process of eastward diversion has been extensively studied from the historical viewpoint, revealing that the initial project in the 17th century was principally intended to establish a stable navigation route. However, no scholars have yet proven this hypothesis via hydrological modeling.

We used the H08 global hydrological model to reconstruct historical flow direction maps at 60-arcsecond spatial resolution with a 1-day temporal resolution. We hypothesized that the historical claims could be numerically verified using a relatively simple simulation. First, we confirmed that our modeling framework reasonably reproduced the present river flows by adding two present-day bifurcation functions. Next, using the reconstructed historical maps, we quantified low flows (20th percentile) in the 17th century and confirmed that the Tone River diversion aided navigation because it connected areas that increased low flows. Finally, the validity of our historical simulation was proven by contrasting the distribution of simulated low flow rates with the flows at the historical river ports that lay furthest upstream. We show that it is possible to bridge two different disciplines, history and numerical hydrological modeling, to obtain a better understanding of human–water interactions. One limitation is that we only reconstructed historical land maps in the present study; the meteorological forcing inputs employed were identical to those of the 20th century; the historical inputs are not known.

**Keywords**: Paleo-hydrological bridge; H08 global hydrological model; Tone River Eastward Diversion; 17th century; maps reconstruction; low flows; navigable paths;





## 1 Introduction

### 1.1 Background


The courses of rivers, particularly those flowing in densely populated areas, are sometimes heavily altered by humans. Such changes are easily recorded if they are recent but not if they happened more than a few centuries ago. One such example is the Tone River, Japan's largest river. The Tone River basin is the Kanto Plain, where the capital, Tokyo, is

located. The Tone River rises in a mountainous region northwest of the Kanto Plain that now ends at the southeastern corner of the plain, where it flows into the Pacific Ocean. Originally, however, the river flowed through only the western half of the Kanto Plain and ended at Tokyo Bay. The eastern half was drained by another river, the Hitachi River, running in the lower reach of the present-day Tone River. Principally during the 17th century (in the Edo Period, see

**Supplementary Text 1**), several remarkable projects connected the two basins of the Tone and the Hitachi. In this study, we term the series of projects the Tone River Eastward Diversion Project (TREDP). The considerable changes in the river courses have attracted the attention of not only hydrologists but also archaeologists and geologists.

### 1.2 Earlier studies


The history of the Tone River has been investigated by numerous researchers over a century. Takashi Kawada wrote the first report on the history of the development of the Tone River (Kawada, 1893). Later, Ryosuke Kurihara wrote a book on the same topic, which became the established view (Kurihara, 1943). In short, both authors claimed that the primary purpose of the 17th century TREDP was to protect the capital Edo (the Tokyo of today, see

**Supplementary Text 2**) from floods. At that time, bypassing of flood water through flood channels was already common in Japan, and was one of the main reasons why the TREDP was conducted (Koide, 1975; Kosuge, 1981). After World War II, Haku Koide and Takashi Okuma





proposed a different view: the primary purpose of the TREDP was to enlarge the navigation

network and enhance low flow to maintain network stability (Koide, 1975; Okuma 1981).

Okuma (1981) hypothesized that the Tone River diversion of the Edo Period greatly enhanced

the sustainable socioeconomic development of the Kanto region. In those days, a substantial

number of people relied on the renewable resources of the Japanese Archipelago but of course

lacked modern technology. Increased low flows and uninterrupted connections between

mountainous and coastal regions expanded the business and trade possibilities, sustaining the

historical supply of food and commodities to the capital Edo. This view is supported by a great

deal of historical evidence and is now regarded as accurate.

### 1.3 The shortcomings of earlier studies

Although the claims of Koide and Okuma have been established from the historical

perspective, little is quantitatively known; ideally, numerical hydrological simulations are

required. To the best of the authors' knowledge, numerical hydrological simulations that

targeted events that took place more than a century ago have seldom been performed,

particularly in Japan. The exceptions include the following studies. Luo et al. (2014) evaluated

the paleo-hydrological anthropogenic impacts of land use changes in the Kamo River basin in

Kyoto, Japan, by reconstructing historical land use maps from 1902. They reported delayed

and reduced peak discharges under the land use regime of 1902 compared to that of 1976

because of decreases in forest and paddy field coverage. Nemoto et al. (2011) developed a

numerical model to investigate the hydraulic effect of the Shingen-tsutumi (a series of dikes

built in the 16[th] century in Yamanashi Prefecture, Japan). They confirmed the hydraulic

efficiency of dikes described in the ancient texts and effectively countered certain recent

alternative views. Nemoto et al. (2013) developed a hydrological model to estimate the

inundation depth at the time of the battle of Bicchu Takamatsu castle in 1582, during which it

is known that the attackers created a temporary earth dike to flood the castle and thus force the





defenders to surrender. It was found that building a relatively short and low structure at a hydrologically critical point was sufficient to inundate the castle; this provided a hydraulic basis for the historical event. Outside Japan, the following studies tackled century-wide historical hydrological reconstruction. Balasch et al. (2019) qualitatively reconstructed and estimated peak flows in different subbasins of the Ebro River basin over the last 400 years. They reconstructed peak flows with possible errors of 10% after calibrating the hydraulic model and found that the geometry of the modeled reach of the riverbed and the flood plain was the greatest source of uncertainty. Werther et al. (2021) reconstructed river channel lengths from the pre-modern period (19th century) to today in the Bavaria region of Germany and calculated changes in the main channels of all major rivers in the study area after the channels were straightened.

### 1.4 Objectives and new contributions

In this study, we model and evaluate the river discharges of the Tone and various other rivers and quantitatively investigate the hydrological consequences thereof during the various stages of the TREDP waterworks, which established more stable navigational routes in the Kanto Plain, including the capital Edo. We use the H08 global hydrological model for this purpose (Hanasaki et al., 2018). This choice was made because global hydrological models are designed for data-sparse regions, and an earlier study indicated that the H08 was capable of simulating Japanese rivers at a reasonably high spatiotemporal resolution (Hanasaki et al., 2022). In particular, the specific questions addressed in this study are summarized as follows:

1) How can we establish a hydrological simulation of events that occurred several centuries ago and that are thus associated with data limitations?

2) How can we validate the simulations for periods when modern river gauging did not exist?



3) What were the implications of TREDP? Were the implications consistent with the views of Koide and Okuma, who claimed that enhancement of navigation was key?

## 2 Study sites

### 2.1 Climatological and hydrological characteristics

The Tone River watershed area is 16,940 km$^2$, and the main channel runs for 322 km from its source to the Pacific Ocean (MLIT, 2023a). The mean annual rainfall in the upper Tone River basin is 1,381 mm with a standard deviation of 148 mm (Ibbitt et al., 2002). The mean Ara, Edo, and Tone discharges (MLIT, 2023b) for the period between 2004 and 2008 at the most downstream stations (Jisui Bashi, Nagareyama, and Toride, respectively) were 103, 111, and 242 m$^3$/s, respectively.

Three rivers, the Tone, the Ara, and the Edo, are connected by two major bifurcations that are too large to ignore. The first bifurcation is the Tone Ozeki Weir, where some Tone River water is diverted to the Ara River (hereafter T2A), and the second bifurcation is the Sekiyado Dam in the Tone River, which controls the amount of water that flows downstream toward the Edo River (hereafter T2E). Presently, the T2E diversion is operated using a gate, whereas there was natural diversion in the 17$^{th}$ century. The only structure was earthen; this narrowed the inlet of the Edo River (termed "Bo-dashi" in Japanese, which means "stick out").

### 2.2 Study sites relevant to the historical Edo Period

As mentioned earlier, TREDP involved numerous subprojects. We focus on six subprojects and reconstruct historical maps of the Tone, Edo, and Ara Rivers in the Kanto region of Japan. The six historical maps were generated electronically by manually changing the flow directions by reference to historical reports on TREDP development, as follows:



before 1593; after checking part-blockage by the Aino River diversion (a bifurcating channel

of the mid-Tone River) in 1594, which is considered to be the first step of TREDP; after

completion of the Shinkawa-Dori channel (straightening of the mid-Tone River) in 1621; after

the Kinu and Kokai River diversions, which moved the confluent points upstream, together

with the Ara River diversion in 1630, which connected the upper basin to a southern river; after

the Akahori River diversion (a new section that connected the upper and lower Tone River) in

1654; and after the Sakasa River diversion and relevant constructions in 1666. Construction of

the Sakasa River with a near-zero slope changed the flow direction from the Tone to the Edo

River and vice versa ("sakasa" translates to "opposite" in English). This was achieved by

narrowing the Edo River; otherwise, all river water drained toward Tokyo Bay, which reduced

the water flowing eastward.

**Figure 1** shows the historical and present-day river basin maps, from 1593 (before the

TREDP) to 2023. All three river basins are located between 35°N and 37.4°N and 138°E to

141°E (i.e., within an area approximately 220 km × 160 km). Using the catchment areas of the

stations farthest downstream that collect discharge data, the three rivers are classified as

follows: Tone (5,002 km$^2$ on historical maps corresponding to the eastern part of the present

Tone River; 11,376 km$^2$ on the present-day map), Edo (7,946 km$^2$ on historical maps

corresponding to the western part of the present-day Tone River; 787 km$^2$ on the present-

day map), and Ara (1,383 km$^2$ on historical maps; 2,168 km$^2$ on the present-day map). On historical

maps, the easternmost river basin was termed the Hitachi River and the middle river basin the

Tone River; they were not connected. However, in the chapters below we use the present-day

nomenclature; the middle river basin is termed the Edo River and the easternmost river basin

the Tone River. Below, we apply this nomenclature to both the present-day and historical maps

for convenience.



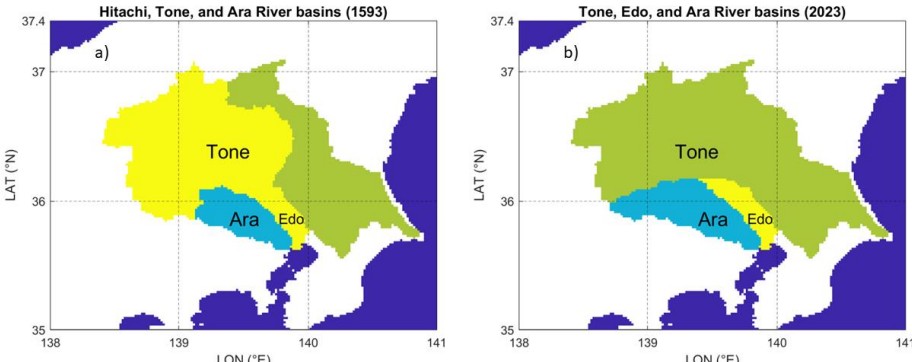

**Figure 1. The Tone (green), Edo (yellow), and Ara (blue) River basins in (a) 1593 and (b) 2023. In 1593, the Tone River mouth corresponded to the 2023 Edo River Mouth, whereas the 2023 Tone River Basin was termed the Hitachi River Basin.**

**Table 1** shows the locations and other information for 20 river gauging stations (MLIT, 2023b), and **Figure 2** presents the river sequences of today with the positions of all 20 gauging

stations relative to the bifurcations.

**Table 1. The present-day gauging locations with observed river discharge. From the left, the columns contain station identification numbers (IDs), station names, the river systems, longitudes, latitudes, watershed areas of the observed data, watershed areas of the H08 model, and errors in the watershed areas of H08. The tributaries in Table 1 are shown later in Figure 2.**


| ID | Station | River | Lon [°E] (obs) | Lat [°N] (obs) | Area [$10^3$ km$^2$] (obs) | Area [$10^3$ km$^2$] (H08) | Areal error [%] (H08/obs) |
|----|---------|-------|-----------|-----------|-----------|-----------|------------|
| 1 | Iwahana | Karasu | 139.08 | 36.29 | 1.23 | 1.39 | +13 |
| 2 | Koga | Watarase | 139.69 | 36.19 | 2.20 | 2.57 | +17 |
| 3 | Mitsukaido | Kinu | 139.98 | 36.02 | 1.74 | 1.94 | +12 |



| 4 | Todai | Kokai | 140.13 | 35.88 | 1.04 | 0.98 | -06 |
| 5 | Kamifukushima | Tone | 139.12 | 36.31 | 3.66 | 3.30 | -10 |
| 6 | Yattajima | Tone | 139.20 | 36.26 | 5.15 | 5.54 | +08 |
| 7 | Furuto | Tone | 139.38 | 36.24 | 5.99 | 6.02 | +01 |
| 8 | Kawamata | Tone | 139.52 | 36.19 | 6.02 | 6.14 | +02 |
| 9 | Kurihashi | Tone | 139.70 | 36.14 | 8.59 | 8.90 | +04 |
| 10 | Tonesekiyado | Tone | 139.77 | 36.12 | 8.56 | 8.96 | +05 |
| 11 | Kitasekiyado | Tone | 139.79 | 36.10 | 8.77 | 8.99 | +03 |
| 12 | Mefukibashi | Tone | 139.89 | 35.98 | 8.85 | 9.37 | +06 |
| 13 | Toride | Tone | 140.06 | 35.89 | 12.2 | 11.5 | -06 |
| 14 | Fukawa | Tone | 140.14 | 35.85 | 12.5 | 12.6 | +01 |
| 15 | Nishisekiyado | Edo | 139.78 | 36.09 | 8.66 | 8.99 | +04 |
| 16 | Noda | Edo | 139.85 | 35.94 | 8.69 | 9.33 | +07 |
| 17 | Nagareyama | Edo | 139.89 | 35.85 | 8.71 | 9.71 | +12 |
| 18 | Uematsubashi | Ara | 139.28 | 36.14 | 0.97 | 0.97 | +00 |
| 19 | Oashibashi | Ara | 139.44 | 36.08 | 1.02 | 1.06 | +04 |
| 20 | Sugama | Iruma (Ara) | 139.51 | 35.95 | 0.71 | 0.82 | +16 |

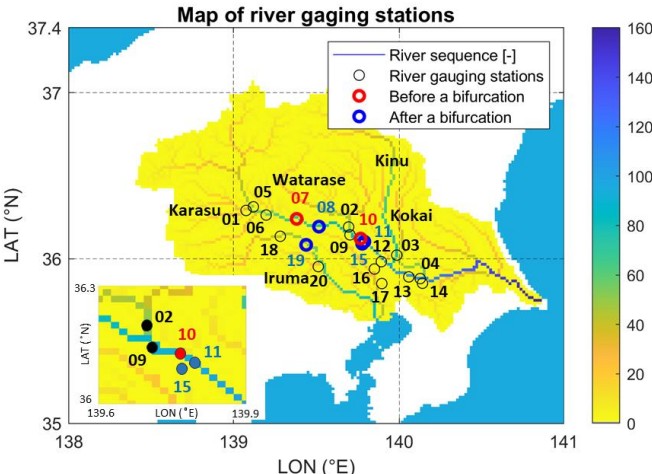

**Figure 2. The present-day river sequences with the positions of all 20 gauging stations relative to the locations of**

**bifurcations. Numbers near stations before (red color) and after (blue color) bifurcations correspond to the station IDs**

**listed in Table 1.**

## 3 Materials and Methods

### 3.1 Overview

The H08 is a grid-based, physically driven hydrological model that calculates the global

hydrological cycle with consideration of major human activities (e.g., sector-wise water

abstraction and reservoir operation) and assesses water resource status at 30 arc-min spatial

resolution (Hanasaki et al., 2008a; Hanasaki et al., 2008b; Hanasaki et al., 2018). The H08 is

applicable to small spatial domains and low resolutions. Recently, the H08 model was

expanded because of the need for regional application in Kyushu Island of Japan; the fine

spatial resolutions are 1 arc-min and a 1-day time step (Hanasaki et al., 2022). We applied the

H08 model to the present-day river map of the Kanto plain, which has the two existing

bifurcation functions T2A and T2E that divert the flow of the Tone River to the Ara and the



Edo Rivers, respectively, at certain fixed rates. Next, we validated the simulated river

discharges of the present-day river map using observed data. Then, the historical river

discharges were estimated by replacing the present-day river map with historical maps. Finally,

we indirectly validated the results by comparing the distributions of low flow and the most

upstream locations of navigable river ports.

### 3.2 H08 model

H08 consists of six submodules, namely land surface hydrology, river routing, reservoir

operation, irrigation water estimation, environmental water estimation, and water withdrawal

submodules. A detailed description, including all fundamental equations together with the

validation results, has appeared elsewhere (Hanasaki et al., 2008a; Hanasaki et al., 2008b;

Hanasaki et al., 2018, Hanasaki et al., 2022). In this study, only the land surface hydrology and

river routing submodules were used. The land surface hydrology submodule calculates the

surface water and energy balance. This submodule requires daily precipitation, air temperature,

wind speed, surface pressure, shortwave and longwave radiation, and relative humidity data as

meteorological forcers. The river routing submodule calculates the daily river discharge by

routing the daily runoff at a constant velocity.

The approach and methodology of Hanasaki et al. (2022) were applied in this study when

using the land surface hydrology and river routing submodules to analyze data from 1993 to

2008. The other submodules were disabled, mainly because their application to historical maps

would render the historical simulations very uncertain given the lack of input data. The

calibration period was 1993–2000, and the validation period was 2001–2008. During this

period, quantitative rainfall and river discharge data were consistently available from all

observation stations.



### 3.3 Data

#### 3.3.1 Meteorological data

The observed meteorological variables were collected from the Automated

Meteorological Data Acquisition System (AMeDAS, 2023) online database of the Japan

Meteorological Agency. These were point data from all 91 rain gauges of the three river basins;

the records have been published and are thus readily available. The modeling framework

formulated for this study is schematically shown in **Figure 3**.

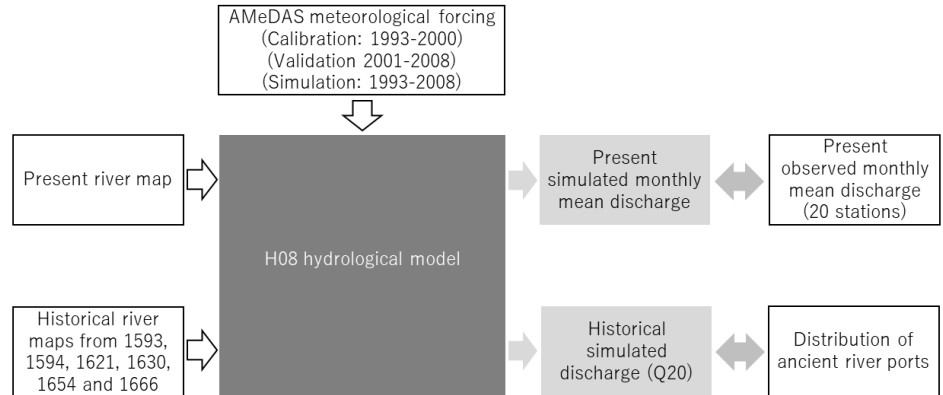

**Figure 3. Schematic of the methodology and framework used in this study.**

#### 3.3.2 River discharge data

The river discharge data were collected from the online database of the Ministry of Land,

Infrastructure, Transport and Tourism (MLIT, 2023b).

#### 3.3.3 Flow direction data

Matsumura et al. (2021) developed a flow direction map for the present-day Tone, Edo,

and Ara River basins with a resolution of 1 arc-min (approximately 2 km). During





reconstruction of historical maps, the flow direction was manually changed at one or more points to match the available historical maps (i.e., MLIT, 2023c; Inazaki et al., 2014). When it was difficult to determine the exact historical river route (the available historical maps are often hand-drawn), both administrative borders and the associated terrain slopes were used as guidelines, because we assumed that these did not change over time even when the river routes varied. The parts of the domain that did not lie inside the three river basins were treated as no-data points.

First, the map before the waterworks started is termed "1593", indicating the year in which the map was prepared. Second, we used the map obtained after part-blockage of the Aino River (1594). Third, we used the maps created after the Shinkawa-Dori diversion (1621), followed by the diversions of the Ara, Kinu, and Kokai Rivers (1630), the Akahori River (1654), and the Sakasa River (1666).

### *3.4 Simulation*

#### *3.4.1 Parameter calibration*

The parameters of the land surface hydrology submodule were tuned using a simple Monte-Carlo-based method that has been described previously (Hanasaki et al., 2022; Mateo et al., 2014; Masood et al., 2015). The sensitivities of four hydrological parameters were evaluated by deriving the Nash–Sutcliffe Efficiency (NSE; **Equation 1**, Nash and Sutcliffe, 1970) of the monthly average river discharges using three possible values of each of the four parameters, for a total of 81 possible parameter combinations (**Table 2**). The calibration period was 1993–2000, and the validation period was 2001–2008. The most frequently occurring best-calibrated parameters from all gauging stations were termed "ensemble optimum parameters" (EOPs). For the validation period, simulated discharges were evaluated and compared to





observations using both the best parameters at each station and the EOPs obtained during the

calibration period with respect to the computed monthly NSE values.

$$NSE = 1 - \frac{\sum_{t=1}^{T}(Q_s^t - Q_o^t)^2}{\sum_{t=1}^{T}(Q_o^t - \overline{Q_0})^2} \tag{1}$$

where $Q_o^t$ is the observed discharge at time $t$, $Q_s^t$ is the simulated discharge at time $t$, and $Q_o$ is

the mean observed discharge during an event.


**Table 2. The four calibrated parameters SD, CD, γ, and τ, and the three possible outcomes A, B, and C.**

| Parameter/Outcome | A | B | C |
|---|---|---|---|
| Soil Depth (SD) [m] | 0.25 | 1.00 | 4.00 |
| Bulk Factor (CD) [-] | 0.002 | 0.006 | 0.010 |
| Shape Factor (γ) [-] | 1.0 | 2.0 | 3.0 |
| Time Constant (τ) [d] | 25 | 100 | 400 |

*3.4.2 Present-day simulation*

The results of three present-day simulations using (a) the default uniform global H08 model

parameters, b) the best-calibrated parameters at each gauging station, and c) the EOPs obtained

during the calibration period were validated using the observed daily discharge values (MLIT,

2023b) from 20 river gauging stations in the present Tone, Edo, and Ara Rivers. Furthermore,

we also conducted sensitivity analyses of both the T2A and T2E operational functions (data

not shown) to determine the constant diversion rates (i.e., the percentages of river discharge

flowing towards each river) when varying the percentages of river discharge flowing toward

the Edo and Tone Rivers. The sensitivity analysis showed that the NSE values for Tone River

observation stations downstream from Tonesekiyado were highest when the diversion rates

were set to 70% of river discharge toward the Tone River and 30% toward the Edo River. To





estimate the flow rate of the T2A diversion, based on the observed river discharge data from the Furuto (upstream, before diversion, $Q_F$) and Kawamata (downstream, after diversion, $Q_K$) stations (**see Fig. 2**), we obtained the following relationship (**Equation 2**).

$$Q_K = 0.1432Q_F + 28.78 \tag{2}$$

This functional relationship is employed to recalculate the river discharge at Kawamata

from the modeled river discharge at Furuto at every simulation step. This relationship was used throughout all present-day river map simulations; it was the operational rule of the T2A diversion.

### 3.4.3 Historical simulation

The default uniform global H08 model parameters and the EOPs from the present-day

maps were selected for the historical simulations of all grid cells when we analyzed historical changes in river discharges. For the historical simulations, the meteorological inputs were the same as for the present-day simulation; we lacked historical meteorological data from the 17[th] century. We applied the same modeling configurations as employed for the present-day simulation, but without T2A and with a modified T2E; the T2E bifurcation function was

adjusted from 70:30% for the present-day to 50:50% for the historical maps (hereafter the old T2E). This was because the flood control of high waters flowing towards the Edo River was not as efficient as is afforded by the Sekiyado Dam of today (Okuma, 1981) and because we assumed that one of the main goals of enhanced navigation through the Sekiyado point was to increase the minimum water levels of both the Tone and Edo Rivers as much as possible and

to equal extents. The bifurcation between the Tone and the Ara River did not exist in the 17[th] century, and the bifurcation between the Tone and Edo Rivers differed from that of today. We assumed that the optimal reproducibility metrics for the old T2E on the historical maps



included 50% flow in each direction, because the discharge of low flows toward each river at the bottleneck crossover of Sekiyado would then be optimal.

*3.5 Analysis*

As historical river discharge records are not available, we indirectly validated our findings using the records of riverine ports. We collected data on the most historically upstream river port locations for each river (Figures with similar data are also in Inazaki et al. (2014), Kubo (2012) and Okuma (1981)). As the widths and depths of historical river routes differed

from those of the present-day, it was challenging to determine the historical thresholds for navigable river routes. We evaluated various historical navigation possibilities under the assumption that navigable river route thresholds would be met when Q20 exceeded 20 m³/s throughout the Kanto plain, and particularly throughout the Hitachi River, which was a bottleneck compromising smooth navigation through the Tone River in the Edo Period. The

validity of the 20 m³/s threshold is discussed below.

**4 Results and Discussion**

In the following, we discuss the results presented in this study as they relate to the three specific questions that were the objectives of the study. Each of the specific questions was

evaluated separately and is discussed in subsections 4.1–4.3, as follows:

1) How can we establish a hydrological simulation of events that occurred several centuries ago and that are thus associated with data limitations?

2) How can we validate the simulations for periods when modern river gauging did not exist?

3) What were the implications of TREDP? Were the implications consistent with the views

of Koide and Okuma, who claimed that enhancement of navigation was key?



### 4.1 Reconstruction of historical Tone River basin maps

As we proceeded through the various steps of the TREDP, we reconstructed the six aforementioned historical maps. **Figure 4** shows the reconstructed river sequences of five historical maps and also the present-day Tone, Edo, and Ara River basins; the color bars

indicate unitless river sequences.

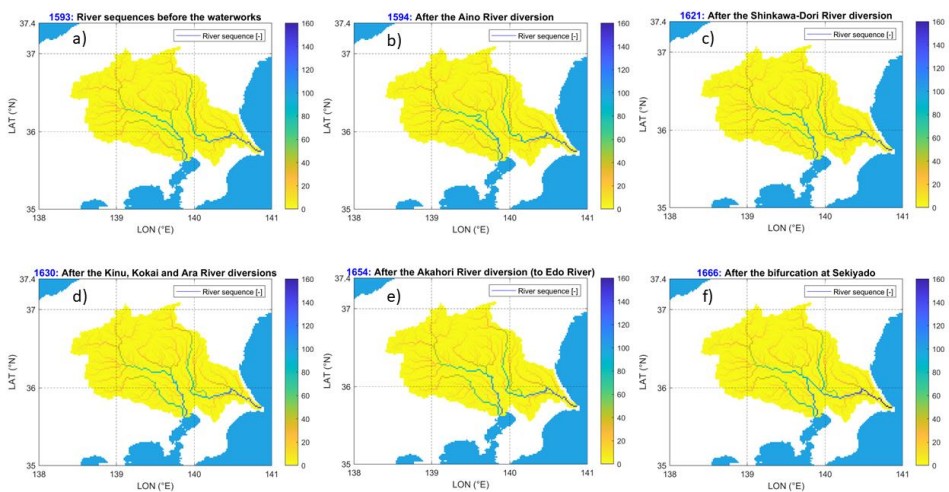

**Figure 4. Reconstructed river sequences [-] of the Tone, Edo, and Ara River routes on five historical maps (a: 1593, b: 1594, c: 1621, d: 1630, e: 1654, f: 1666 map).**

The river sequences after the old T2E bifurcation of 1666 that sent 50% of the water

flow toward the Edo and Tone Rivers at the Sekiyado point are visualized in **Figure 4f**. The historical maps reconstructed from the several individual steps of the TREDP enhance our understanding of engineering technologies that were well-understood and applied even in the 17th century. Particularly, the 17th century engineers planned TREDP step-by-step, and thus gradually moved the Tone River mouth to a distance about 100 km from the original mouth.




### *4.2 Calibration and validation: Present-day Tone River discharge*

#### *4.2.1 Tone River to Ara River (T2A)*

The relationship between the discharge at Furuto (before T2A; see **Fig. 2** and **Table 1**) and Kawamata (after T2A) stations from 1993 to 2008 is shown in **Figure 5**. After removing

the five outlier data points, we obtained **Equation (2)**, as introduced in chapter 3.4.2.

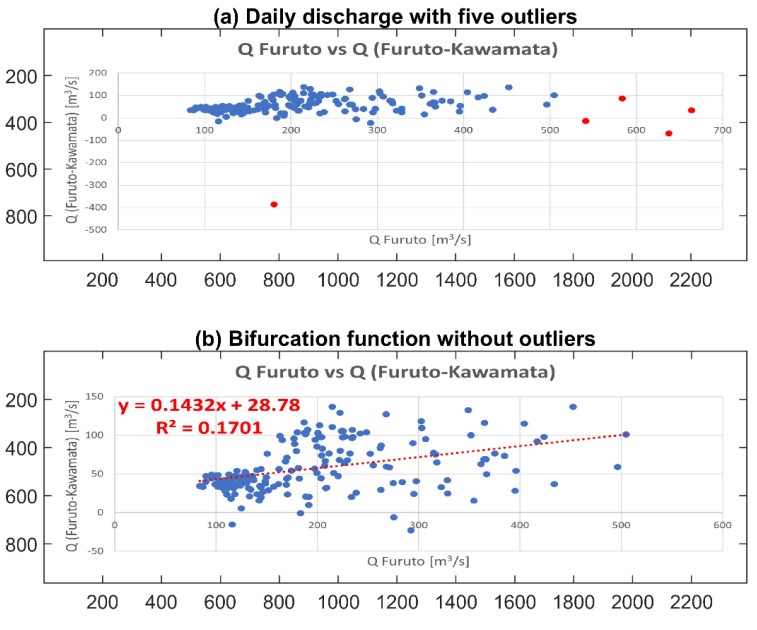

**Figure 5. The process used to obtain the observed discharge correlation function between (a) Furuto and (b) Kawamata stations, as formulated by Equation (2).**

**Figure 6** shows the best NSE reproducibility metrics at the Kawamata and Tonesekiyado gauging stations for the monthly discharge simulations from 1993 to 2008.



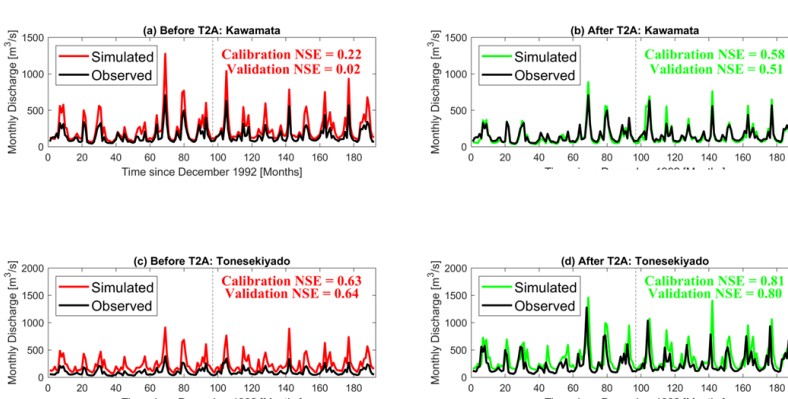

**Figure 6. The best NSE reproducibility metrics at the Kawamata (upper panels) and Tonesekiyado (lower panels) gauging stations before (a, c) and after (b, d) implementation of the T2A bifurcation, for both the calibration and**

**validation periods.**

The monthly NSE value at Kawamata station was only 0.22 for the calibration and 0.02 for the validation period when the bifurcation function was not applied between the Furuto and Kawamata stations at the point of the Tone Ozeki Weir. We obtained the relationships at Furuto

and Kawamata from the observed data, as explained in section 3.3.2. After implementing the T2A bifurcation function mentioned above, the monthly NSE value at Kawamata station greatly increased from 0.22 for the calibration and 0.02 for the validation period (**Fig. 6a**) to 0.58 and 0.51 (**Fig. 6b**) and, further downstream at Tonesekiyado station, increased from 0.63 and 0.64 (**Fig. 6c**) to 0.81 and 0.80 (**Fig. 6d**), respectively. Consequently, the reproducibility

metrics of all downstream stations also significantly increased (data not shown).





*4.2.2 Tone River to Edo River (T2E)*

**Figure 7** shows the best NSE reproducibility metrics at the Kitasekiyado and Nishisekiyado gauging stations for the monthly discharge simulations from 1993 to 2008 after the T2A and T2E bifurcations. Tonesekiyado station is located upstream from T2E, followed

by the downstream stations Kitasekiyado in the Tone River and Nishisekiyado in the Edo River (see **Fig. 2;** zoomed region in the lower left corner).

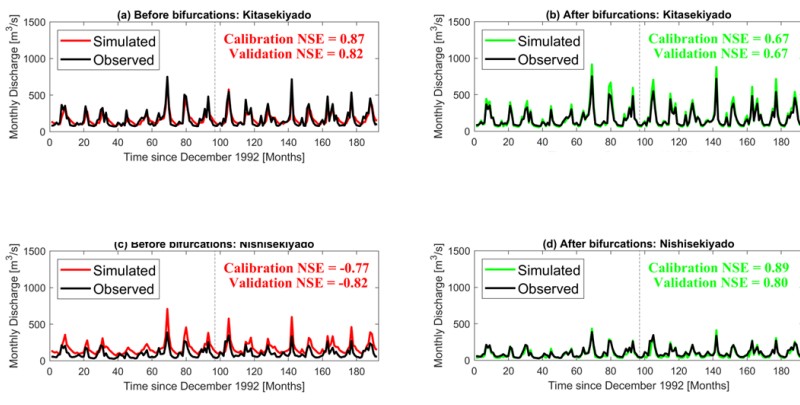

**Figure 7. The best NSE reproducibility metrics at the Kitasekiyado (upper panels, after T2E toward Tone) and Nishisekiyado (lower panels, after T2E toward Edo) stations before (a, c) and after (b, d) implementation of the T2A**

**and T2E bifurcations, for both the calibration and validation periods.**

As shown in **Figure 7**, the monthly NSE value at Kitasekiyado station, located just downstream from T2E, decreased from 0.87 for the calibration and 0.82 for the validation period to 0.67 and 0.67, respectively, when using the best combination of calibrated parameters during implementation of the T2A function into our H08 modeling system. This shows the

impact 30 km downstream of the T2A bifurcation. However, the validity at Nishisekiyado station greatly increased from -0.77 for the calibration and -0.82 for the validation period to



0.89 and 0.80, respectively. The rationale for choosing 70% flow toward the Tone River and 30% flow toward the Edo River at the T2E bifurcation was that the Kitasekiyado and Nishisekiyado stations should exhibit near-equal NSE metrics after implementation of the bifurcation, which was achieved.

*4.2.3 Validation at stations*

**Table 3** shows the calibration and validation data for the modeled monthly river discharges using the observed river discharges from 20 gauging stations and their EOPs for the present-day map.






**Table 3. Calibration (from 1993 to 2000) and validation (from 2001 to 2008) of the modeled monthly river discharges based on the observed river discharges from 20 gauging stations for which NSEs were derived using the best parameters and the EOPs for the present-day map. The columns (from left) are the station IDs, the optimum parameters for the calibration period, the best NSEs for the calibration period, the best**
**NSEs for the validation period, the NSEs of the EOPs for the calibration period, and the NSEs of the EOPs for the validation period.**

| Station ID | Calibration period best parameters combination | Calibration period best parameters NSE | Validation period best parameters NSE | Calibration period EOPs NSE | Validation period EOPs NSE |
|---|---|---|---|---|---|
| 1 | CAAC | 0.56 | 0.62 | 0.56 | 0.58 |
| 2 | CACB | 0.35 | 0.82 | 0.33 | 0.81 |
| 3 | CBAC | 0.87 | 0.87 | 0.82 | 0.77 |
| 4 | CCCC | 0.74 | 0.52 | 0.64 | 0.47 |
| 5 | CACB | 0.24 | 0.01 | 0.24 | -0.14 |
| 6 | CAAC | 0.39 | 0.18 | 0.39 | 0.04 |
| 7 | CAAC | 0.58 | 0.39 | 0.58 | 0.35 |
| 8 | BACA | 0.58 | 0.51 | 0.58 | 0.38 |
| 9 | CAAC | 0.81 | 0.81 | 0.81 | 0.80 |
| 10 | CAAC | 0.81 | 0.80 | 0.81 | 0.79 |
| 11 | CAAC | 0.67 | 0.67 | 0.67 | 0.67 |
| 12 | CAAC | 0.61 | 0.51 | 0.61 | 0.50 |
| 13 | CAAC | 0.66 | 0.79 | 0.66 | 0.75 |
| 14 | CAAC | 0.55 | 0.75 | 0.55 | 0.71 |
| 15 | BACA | 0.89 | 0.80 | 0.88 | 0.76 |
| 16 | CAAC | 0.90 | 0.90 | 0.90 | 0.88 |
| 17 | BACA | 0.70 | 0.71 | 0.66 | 0.69 |
| 18 | BABA | 0.22 | 0.27 | 0.19 | 0.26 |
| 19 | BACA | 0.58 | 0.55 | 0.58 | 0.54 |
| 20 | CACC | 0.77 | 0.64 | 0.76 | 0.64 |
| Ensemble optimum parameters | **CAAC** | **0.62** | **0.60** | **0.61** | **0.56** |

**Table 3** provides an overview of the simulation results; it is apparent that 16 of 20 stations had NSE values > 0.50 during both the calibration and validation periods. The stations

that exhibited the lowest NSE values in both the calibration and validation periods were the most upstream stations on the Tone (Kamifukushima, Yattajima) and Ara (Uematsubashi)



Rivers. This may indicate that these relatively smaller subbasins may require different optimal combinations of parameters than the combinations that we selected. However, as the downstream parts of all three rivers yielded relatively high reproducibility metrics, we selected

CABC to be the set of EOPs (see **Tables 2** and **3**) when further analyzing river discharges. The mean NSE values of all 20 stations obtained using the best parameters for every station were 0.62 for the calibration period and 0.60 for the validation period, which are satisfactorily high values. When the EOPs were applied to the entire basin over the calibration and validation periods, the obtained mean NSE values were 0.61 and 0.56, respectively. The very small drops

in the NSE reproducibility metrics from the cases with the best NSE values for each station to the cases with the NSE values obtained using the EOPs indicate that it is appropriate to use the CABC EOPs for the entire basin.

*4.2.4 Basin-wise water balance*

**Table 4** analyzes and compares the present-day simulated and observed data on the

mean annual simulated discharges from 1993 to 2008 at the Tone, Edo, and Ara River mouths using the default uniform global H08 model parameters when (a) both the T2A and T2E bifurcation functions were operating simultaneously and when (b) only the T2E bifurcation was operating.






**Table 4. Mean annual simulated discharges at the mouths of three rivers in the present-day (m³s⁻¹) using the default uniform global H08 model parameters.**

|  | Present Simulation including both T2A and T2E bifurcations | | Present Simulation including only the T2E bifurcation |
| --- | --- | --- | --- |
|  | Sim | Obs | Sim |
| Tone | 303 | 242 | 332 |
| Edo | 76 | 111 | 89 |
| Ara | 104 | 103 | 62 |
| T2A | 42 |  | 0 |
| T2E | 54 |  | 67 |

**Table 4** shows the obtained present-day mean discharges at the mouths of the Tone,

Edo, and Ara Rivers. These are 303, 76, and 104 m³/s, respectively, when the T2A and T2E

bifurcations operate. The Ara River value is similar to the 2004–2008 mean annual observed

discharge (MLIT, 2023b) of the Ara River (103 m³/s), but the Edo River discharge is

underestimated and the Tone River discharge is overestimated compared to the observed values

(111 and 242 m³/s, respectively). Without the T2A bifurcation, the errors are greater, as

expected. Importantly, the observed discharges were those at the most downstream stations but

the simulated discharges were those at the mouths; it would thus be expected that the simulated

discharges would be slightly larger. The mean annual discharges of the Tone River are 151 and



101 m$^3$/s before and after the T2A point, and those of the Ara River are 24 and 66 m$^3$/s, respectively. Thus, about 42 m$^3$/s is diverted by T2A on average.

420        Summarizing **Figures 6 and 7** and **Tables 3 and 4**, we conclude that our H08 modeling framework with T2A and T2E accurately reproduces the present-day Tone, Edo, and Ara River discharges, especially in the most downstream parts.

### 4.3 Quantification of increasing low flows during and after the waterworks

In the following, we quantify the low flows of the Tone and Edo Rivers that were altered
by the TREDP. The need to increase low-flow water levels along the Tone and Edo River systems, particularly in the unnavigable upmost section of the Hitachi River (presently between the Mefukibashi and Tonesekiyado stations of the Tone River; see **Table 1**), has been assumed to be the main reason why the TREDP was conducted and proceeded (Koide, 1975; Okuma, 1981). Note that the capital Edo was also protected from floods by these waterworks (Sippel,
2014). One constraint imposed on flood protection was the Akahori River, which was artificially constructed in 1654 and only 18 m wide at the time of construction (Okuma, 1981) but was subsequently further widened in multiple steps for decades and centuries after the TREDP; however, high flows are not the topic of the present study.

**Figure 8** shows the 20$^{th}$ percentiles (Q20) of monthly river discharges from 1993 to
2008 using the default uniform global H08 model parameters, which served as the reference level when evaluating low flow conditions. Additionally, **Figure 8** also shows the most upstream navigable historical river ports with their Q20 thresholds for 1593, 1630, and 1666 (panels a–c) and compares these to those of all recorded navigable historical ports (panel d).



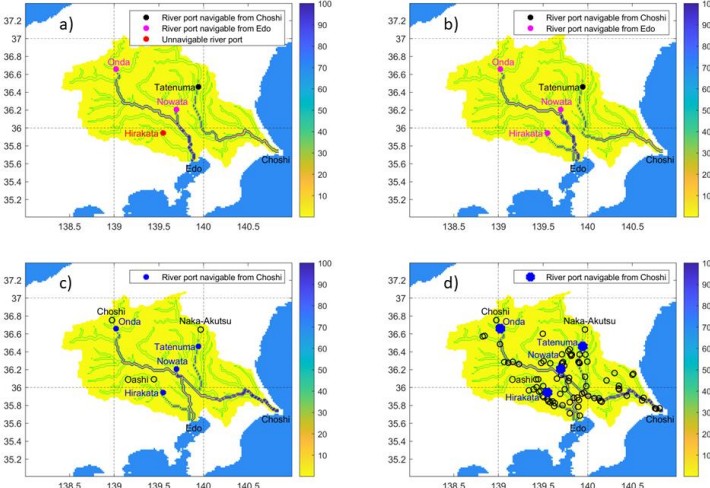


**Figure 8. The contours of Q20 values > 20 m$^3$/s for the historical maps from (a) 1593, (b) 1630, (c) 1666 with the locations of the most upstream navigable historical river ports and (d) all recorded navigable historical ports using the default uniform global H08 model parameters. Historically, there were two Choshi ports, one at the river mouth and the other upstream of the Tone River.**

445       The key confirmation of the expectation enunciated by Koide (1975) is shown in **Figure 8**, which is an indirect validation of the historical maps. **Figure 8** shows our H08-modeled Q20 values and compares these with those of the river ports, whose most upstream locations with Q20 values > 20 m$^3$/s are shown. Q20 values > 20 m$^3$/s indicate navigable points. Note that, in 1593, the eastern half of the present-day Tone River (the Hitachi River) and the western half

of the Tone River were not connected and thus were unnavigable (**Fig. 8a**). Moreover, the Q20 of the Hitachi River exceeded 20 m$^3$/s only after the confluence of the Kinu and Kokai Rivers. In 1630, after diversion of the Kinu River, the section wherein Q20 exceeded 20 m$^3$/s moved upstream (**Fig. 8b**). After the Sekiyado bifurcation (i.e., the old T2E) in 1666, the navigable section was substantially extended, now connecting the upper streams of major tributaries to

the Pacific Ocean and Tokyo Bay (**Fig. 8c**). Furthermore, the distances between the most navigable river ports modeled in **Figure 8c** and the most upstream navigable historical river





ports (i.e., Noda and Choshi for the Tone River; Tatenuma and Naka-Akutsu for the Kinu River; Hirakata and Oashi for the Ara River) are rather small. Thus, we indirectly validated the modeled locations of the uppermost reachable ports of the mainstream Tone and its tributaries

via comparisons with the reconstructed historical uppermost ports.

**Figure 9** shows the validations of the Tone River discharges at Toride station for 3 key historical years, thus 1593, 1630, and 1666, compared to the observed data from 1993 to 2008 derived using the EOPs. For these historical validations, we used the old T2E historical bifurcation ratio at Sekiyado.

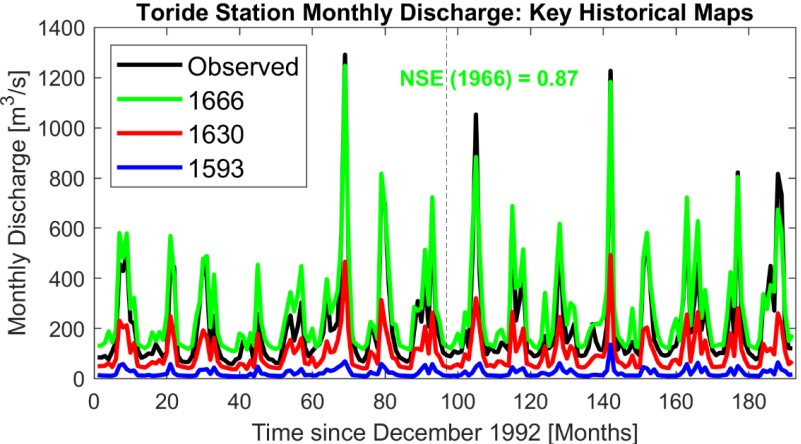


**Figure 9. The Tone River discharges at Toride station in 1593 (blue), 1630 (red), and 1666 (green) compared to the observed data from 1993 to 2008 derived using the EOPs and the old T2E historical bifurcation ratio at Sekiyado.**

The aforementioned increases in low flows are shown in **Figure 9** for Toride station, and the modeled river discharges are compared to the observed values of the present-day

observed values, because these yielded the best reproducibility metrics over all river basins. A



gradual historical increase of Q20 at Toride station is apparent; the values were 11, 53, and 131 m$^3$/s for the maps of 1594, 1630, and 1666, respectively. These findings confirm the expectation of Koide (1975) that the main goal of the TREDP was a step-by-step increase in low flows to enhance navigation.

Summarizing **Figures 8 and 9**, the TREDP was useful in that it revealed increasing low flows over time in the Tone River between Sekiyado and Choshi. TREDP principally used human power to build earthen and wooden structures that allow only limited water diversion. In the early 20$^{th}$ century, modern steel and concrete structures permitted large-scale water transfer during severe floods. Inland transportation has gradually lost importance given the

development of a sophisticated rail network in the region. It is thus not immediately intuitive to assume that transport was the main reason for the TREDP when viewing the scenario today. The socio-hydrological implications and societal contributions of the TREDP expanded the business and trade of the Kanto Plain, which historically supplied food and commodities to the capital Edo. The results support the claims of Haku Koide and Takashi Okuma: the TREDP

was indeed intended to expand the riverine transport network of the entire Kanto Plain.

### 4.4 Uncertainties

Here and below, we note some important limitations of our study. First, the H08 historical maps were forced using present-day meteorological data. The flows may have differed from what we derived because the climatological conditions may have varied.

Recently, a few studies (i.e., Hatono et al., 2022) have developed long-term historical meteorological datasets for Japan, which should enable this limitation to be minimized in future studies. In the cited work, the period from 1926 to 2020 was reconstructed. Such long-term meteorological reanalyses are needed to extend input precipitation data coverage further into the past. Second, when tuning the present-day climate parameters, the effects of dams were not

included, and the subbasins of tributaries were not calibrated separately. Instead, the best-




calibrated parameters for each gauging station were validated using the observed daily discharge values and then applied uniformly throughout the entire basin. Third, water withdrawal and consumption were not included in the simulation. This might affect the water balance in both the middle reaches, where irrigation is concentrated, and the lower reaches

where municipalities are concentrated. Fourth, changes in historical land use were not considered because available historical data were lacking. Fifth, given the lack of data, we did not consider river channel cross-sections. Although navigability is primarily determined by the water depth, width, and flow velocity, we considered only the flow rate for simplicity.

**5 Conclusions**

In this study, we asked and answered the three questions: how can we establish a hydrological simulation of events that occurred several centuries ago and that are thus associated with data limitations?; how can we validate the simulations for periods when modern river gauging did not exist?; and what were the implications of TREDP? Moreover, were the implications

consistent with the views of Koide and Okuma, who claimed that enhancement of navigation was key?

First, we modeled the Tone, Edo, and Ara River discharges before and after the various steps of TREDP and evaluated the results of each step (**Fig. 4**). We used the H08 global hydrological model to reconstruct historical flow-direction maps and determined the

operational patterns of the two present-day bifurcation functions, T2A and T2E. Thus, we successfully established a hydrological simulation commencing four centuries ago.

Second, we found that the navigability paths for low flows (Q20) became connected between the Pacific Ocean and the capital Edo after the TREDP waterworks were completed



(**Fig. 8**) and that the river discharges at the bottleneck Toride station were gradually increased

after conducting each significant waterwork of TREDP (**Fig. 9**).

Third, the societal implications of the diversion afforded numerous advantages, as previously indicated by other scholars. Our simulation results show that the increasing low-flow water levels and thus the improved navigability routes were the most significant societal implication. Thus, our findings are consistent with those of the historians Koide and Okuma.

This is the first study to show that securing stable and adequate low riverine flows permitted navigation, the transport of large amounts of goods from the eastern Kanto Plain to inland areas, and ultimately to the capital Edo, which became safer and economically more efficient than with combined sea navigation and land transportation; we used a historical hydrological modeling framework to discover this.

Learning from historical technologies, followed by efforts to implement what is learnt in the present-day, may be an important step toward more sustainable present-day and future water resources management. We expect that the findings of this study will motivate the scientific community to consider that learning from the past is no less important than projecting and predicting the future, where such learning may yield valuable tools aiding present-day and

future climate studies. For example, in this study, we learnt that low-flow but navigable river routes were designed in a very organized manner even four centuries ago.\

**Code and data availability**

The code and data associated with this study can be accessed at https://doi.org/10.5281/zenodo.10719388.

**Supplementary material**

Supplementary material related to this article is available online.



**Author contributions**

Both co-authors reviewed, discussed, and suggested revisions for the submitted version of the manuscript, conceptualized and designed the experiments, which were then conducted by JT.

NH developed the model code and JT further improved it for the particular application and performed the simulations. JT wrote the manuscript with contributions from NH.

**Competing interests**

The corresponding author declares that none of the co-authors have any competing interests.

**Acknowledgments**

This study was supported by the Japan Society for the Promotion of Science (KAKENHI; grant no. 21H05178). The authors sincerely thank Akiko Matsumura for providing flow direction maps of the present-day Tone, Edo, and Ara Rivers; Kinuyo Tarumoto for digitizing the historical river port locations; and Kedar Otta and Ai Zhipin for their useful suggestions in the early phases of H08 model framework development.

**Financial support**

This study was supported by the Japan Society for the Promotion of Science (KAKENHI; Grant Number 21H05178).

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
