# Peer review of "Simulating the Tone River Eastward Diversion Project in Japan Carried Out Four Centuries Ago"

_EGUsphere, 2024_

## Author Comment (AC1)

**RC1**: 'Comment on egusphere-2024-595', Anonymous Referee #1, 13 Jun 2024

RC1-0. This is interesting study because it quantitatively verified Japan's historic river diversion project using a high-resolution global hydrological model and historic data developments. Reconstructing hydrological conditions from centuries ago, when data were scarce, is challenging and contributes not only to validating specific historical events but also to advancing hydrological analysis.

**Thank you for your useful evaluation and comments, which will surely improve the overall quality of our manuscript. Below are our replies to your three comments. We hereby hope that you will accept our rationale in your comments and approve the changes that we will make based on your useful suggestions. Should some of the comments still be insufficiently addressed, we will gladly address them in detail at the next revision round based on your further suggestions.**

**We will reply to the referee comments, with our answer indicated by Answer (in green), corresponding actions indicated by Action (in blue), and textual changes in *italic font*:**

RC1-1. One objective of this study is to test the hypothesis from previous studies that the diversion project's purpose was to enhance low flows to maintain the stability of the navigation network. References to other English literature on this project have indicated various interpretations of its purpose, including land reclamation, military defense, and flood protection (e.g., Mushiake, 1988). Although this study tested one of these interpretations, it is questionable whether it is appropriate to draw conclusions about such an important aspect of river history based solely on the results of this study, which focused on a single interpretation. While the quantitative validation is significant, conclusions should be considered with verification for other interpretations of the project.

**Our Google searches "Mushiake (or Musiake)", "1988", and "Tone" in various combinations did not show any relevant results in the first 100 entries, so we cannot directly reply to that citation without the exact reference provided. Hereby, we assume that the citation mentions various reasons for conducting the TREDP, as indicated in your comment. We agree with you that there are/were many interpretations of the project.**

**As for your indication that we might oversimplify the complex background, we take it with gratitude. In revision, we will carefully go through the draft again and modify any oversimplification that was made. Also, we will mention that there are more interpretations regarding the project and reflect this fact in conclusions.**

**Regarding the latter half of the RC1-1 comment, while we agree that other reasons might also be as important as enhanced river routes for navigation and trading, a very different modelling setup would be needed to test and show the other hypotheses. Testing them would also be a very interesting contribution to a better understanding of the historical hydrology of the rivers in the Kanto Plain, but apparently, they are out of the intended scope of this study.**

RC1-2. The bifurcation function is adjusted from 70:30 in the present day to 50:50 in the historical figure, but the validity of this adjustment is unclear. This bifurcation function is crucial in determining the low flow of the divergent rivers. There is a risk that the stability of the low flow/navigation network of past divergent rivers is almost entirely determined by this function. When examining historical events, making such a bold assumption about this critical figure is questionable. At least some cases of this function need to be verified.

**Thank you for this comment. We agree with your point that this is an important assumption in our study. First, as we demonstrated in the text, the rate of 70:30 in the present day is reasonable from the viewpoint of river discharge time-series data analysis. As for river discharge 400 years ago, it is hard to know it because there was no quantitative observation left. One thing for sure is that the channel capacity of the Akahori River section, which connects the Edo River (the original southward route) and the Hitachi River (the new eastward route), was much smaller than today; hence most likely less than 70% of river water can travel the Akahori River. We will further justify the rationale of the rate in the revised text. Please note that a relevant sensitivity test has already been conducted, and we confirm that it does not influence our overall conclusions.**

RC1-3. The historical river port locations are used as validation data, but the nature, validity, and reliability of this historical data need to be clarified. As the authors indicate, one of the critical contributions and challenges of this study is validating the simulation results in an era without modern river measurements. The reliability of this validation data could determine the significance of this study. Additionally, the process of developing historical data of river channels should be explained in more detail.

**Thank you for this suggestion. The nature, validity, and reliability of port location data will be further disclosed in the revised text. In short, the names of ports were taken from historians' books and confirmed by the name of "aza", an administrative designation of small sections into which some of the rural districts of Japan are divided. The old river routes were also taken from figures in books and digitalized to maintain the geometry of routes.**

**[Nature] The nature of the presented data is from the collective sources of historians, geographers, and other types of researchers who collected and reported those data. We carefully merged these resources and reconstructed our river port locations based on their reports.**

**[Validity] The validity of our presented river ports data arises from validations of the sources (i.e., ancient documents) who first reported them. Yet, some of them can be validated nowadays using geographical GIS and the name of "aza" mentioned above.**

**[Reliability] The reliability of our presented river ports data was tested by crosschecking several sources (i.e., books and papers) that reported similar data.**

**We agree that the process of developing historical data on river channels should be explained in more detail, we will provide clear explanations to precisely clarify the applied reconstruction processes of the historical maps.**

---

## Author Comment (AC2)

**RC2:** ['Comment on egusphere-2024-595'](), Anonymous Referee #2, 13 Aug 2024

RC2-0. This manuscript presents an interesting study attempting to simulate the streamflow in a hydraulic engineering project influenced basin several centuries ago, and illustrate the values of Tone River Eastward Diversion Project in aiding navigation. The reconstructed streamflow is indirectly validated by some proxies, but no datasets of ancient hydrological and meteorological datasets are used to force or validate the model, which forms a significant disadvantage for a streamflow reconstruction study. However, I think this would be a small issue if the research topic can be adjusted slightly. Besides, I find some important methods/results are presented inadequately or unclearly. Consequently, I recommend a major revision before publication, by addressing the following major and specific comments.

> **Thank you for your useful evaluation and constructive suggestions. We hereby hope that you will find our response satisfactory. If there are some points that are still unclear or require further clarification, we will gladly address them with carefulness in the following round of revision.**
>
> **We will reply to the referee comments, with our answer indicated by Answer (in green), corresponding actions indicated by Action (in blue), and textual changes in *italic font*:**

RC2-1. Expression of the research focus

When talking about historical streamflow reconstruction, we would certainly expect the reconstruction of the climate factors, since climate forcing is one of the most important factors influencing streamflow, leading to strong hydrological nonstationarity. This study mainly explores the streamflow produced by different historical maps. Although authors have acknowledged the limitation in this aspect, I still think the lack of climate reconstruction is an inevitable shortage. However, if we regard this study as an investigation on the influence of TREDP on streamflow, the lack of climate reconstruction would not be so important. Consequently, I suggest the authors to change the expression of the research topic, focusing on the influence of TREDP, rather than "reconstructing" the "historical" streamflow.

> **We appreciate your comment. We understand the word "reconstruct" infers a quantitative estimation of the past. We will carefully revise the entire draft and make sure that the word will be only used for river maps. We will further clarify that streamflow reconstruction is not achieved because of the lack of historical climate data.**
>
> **Your suggestion is actually already included in research question (3) of our study, which states: „What were the implications of TREDP? Were the implications consistent with the views of Koide and Okuma, who claimed that enhancement of navigation was key?"**

RC2-2. Model calibration

RC2-2-1. I am confused about the calibration procedure. The authors only adopt three different values for each procedure, totally 81 combinations. This seems not a normal calibration procedure, in which simulation discharge produced by a large number of parameters need to be calculated.

We agree with you that the calibration procedure is different from many other models. H08, including surface energy balance calculation, has long adopted this procedure and demonstrated it worked for many large basins (e.g., the Chao Phraya River in Thailand reported by Hanasaki et al., 2014; The Ganges Brahmaputra Meghna Rivers by Masood et al., 2015; The rivers in Kyushu island in Japan by Hanasaki et al., 2022). There are four calibratable parameters and minimal three optional values (the minimum, mean, and maximum values of physical bounds for each parameter). Optional values can be increased (e.g., Masood et al. 2015 used 5 values for each). Due to the fact that quite a limited number of parameters were tested, the parameter set found may not be the mathematical optimum, while the parameter set produces reasonable hydrological simulation in the entire basin (see text).

RC2-2-2. Besides, the T2E bifurcation function is set as 70:30% because the NSE values are highest at this time (L262-264). Is NSE always highest at 70:30% for all the parameter combinations? Several lines afterward, the authors claim that the rationale for this value is the near-equal NSE of Kitasekiyado and Nishisekiyado stations. Is this consistent with the sensitivity analysis described previously?

For your first question, our answer is no. NSE was highest at 70:30% only for the best parameter combinations after the T2E bifurcation (see Figures 7b and 7d; the same NSE data are shown in Table 3 on the third and the fourth columns). For your second question, our answer is yes. We tested various combinations of the discharge ratio at the T2E bifurcation, and we selected the value of 70:30%, which exhibited near-equal NSE for Kitasekiyado (after T2E bifurcation, the eastern route) and Nishisekiyado (the southern route) stations for the best parameter combinations (NSE = 0.67 for both calibration and validation at Kitasekiyado and as NSE = 0.89 for calibration and 0.80 for validation at Nishisekiyado). The near-equal NSE at two stations implies a reasonable bifurcation over time.

RC2-2-3. Shouldn't this bifurcation function be determined by something like project planning?

Thank you for this suggestion. There must be a strict gate operation rule at the Sekiyado bifurcation point (so far, the authors have not yet accessed to the written rule yet), but it doesn't necessarily be the solution because the simulation includes bias and errors. Rather important is that the flow simulation at the upper and lower stations agrees well with the observation. We will further revise the text to clarify our intention.

RC2-3. Method description

The reconstruction of the historical maps seems to be a very important part in this study, because the simulation of historical streamflow is produced by simply replace the present map by historical ones. However, the method producing historical maps is only described by several simple sentences and referring to several literatures. I suggest authors to describe this part in details.

We described the methodology for reconstructing the historical maps in section 3.3.3 (Flow direction data), but we agree that the explanation is insufficient. For instance, we stated that "administrative borders and the

**associated terrain slopes were used as guidelines because we assumed that these did not change over time even when the river routes varied"**, but we did not give a particular example of when this was used (i.e., Kinu and Kokai Rivers). In the revised manuscript, we will enhance explanations about methods used for reconstructing historical maps in very much detail, to the point of high methodological reproducibility.

RC2-4. Figure 8 is an important figure showing the validation of historical simulation, but I find this figure and the interpretation on it extremely difficult to follow. What is the meaning and unit of the colorbar? What do the hollow circles represent? What do the channels with deep and shallow color in each map mean? The units of frame should be added. Also, I suggest the authors to refer to the specific point/line in the figure (e.g., … as shown by the XX point in Figure X) when describing this figure in the main text.

We agree that Figure 8 was difficult to follow. Below are replies to all questions.

The main channel indicates river flow meets the condition of Q20 > 20 $m^3$/s, which increases over time.

The color bar indicates the river discharge of the basin; it will be replotted differently in the revision.

The hollow circles represent all recorded navigable historical ports except the most upstream navigable ones. They are only in Figure (8d) to not have too many circles overlapping.

Channels with pale color represent river flow slightly bigger than Q20 > 20 $m^3$/s while the ones with dark blue color indicate bigger discharge up to values in the color bar.

We will add LAT/LON units to the frame.

We will modify the manuscript and Figure 8 by properly specifying figures/panels/points wherever and whenever applicable.

RC2-5. -Figure 6: Maybe replace "before" and "after" by "without" and "with". "Before" and "after" seem to describe the time when an event happens, which may lead to misunderstanding.

Thank you, we agreed and will correct this both for Figure 6 and Figure 7.

RC2-6. -L461: I think Figure 9 is just showing the simulated streamflow of different historical years, not a "validation".

Thank you, we agreed and will correct this.

RC2-7. -Figure 9: 1966 should be 1666?

Thank you, we agreed and will correct this.

---

## Author Response (AR1)

**RC1**: ['Comment on egusphere-2024-595'](), Anonymous Referee #1, 13 Jun 2024

RC1-0. This is interesting study because it quantitatively verified Japan's historic river diversion project using a high-resolution global hydrological model and historic data developments. Reconstructing hydrological conditions from centuries ago, when data were scarce, is challenging and contributes not only to validating specific historical events but also to advancing hydrological analysis.

> **Thank you for your useful evaluation and comments, which surely improved the overall quality of our manuscript. Below are our replies to your three comments. We hereby hope that you will accept our rationale in your comments and approve the changes that we made based on your useful suggestions. Should some of the comments still be insufficiently addressed, we will gladly address them in detail at the next revision round based on your further suggestions.**
>
> **We replied to the referee comments, with our answer indicated by Answer (in green), corresponding actions indicated by Action (in blue), and textual changes in *italic font*:**

RC1-1. One objective of this study is to test the hypothesis from previous studies that the diversion project's purpose was to enhance low flows to maintain the stability of the navigation network. References to other English literature on this project have indicated various interpretations of its purpose, including land reclamation, military defense, and flood protection (e.g., Mushiake, 1988). Although this study tested one of these interpretations, it is questionable whether it is appropriate to draw conclusions about such an important aspect of river history based solely on the results of this study, which focused on a single interpretation. While the quantitative validation is significant, conclusions should be considered with verification for other interpretations of the project.

> **Our Google searches "Mushiake (or Musiake)", "1988", and "Tone" in various combinations did not show any relevant results in the first 100 entries, so we cannot directly reply to that citation without the exact reference provided. Hereby, we assume that the citation mentions various reasons for conducting the TREDP, as indicated in your comment. We agree with you that there are/were many interpretations of the project.**
>
> **As for your indication that we might oversimplify the complex background, we take it with gratitude. In the revision, we carefully went through the draft again and modify any oversimplification that was made. Also, we mentioned that there are more interpretations regarding the project and reflect this fact in conclusions.  We added the following text in the revision:**
>
> **Discussions***: Other reasons might also be as important as enhanced river routes for navigation and trading, but a very different modelling setup would be needed to test and show the other hypotheses (e.g., irrigation area expansion, flood protection, military purpose, etc.). Testing them would also be a very interesting contribution to a better understanding of the historical hydrology of the rivers in the Kanto Plain, but apparently, they are out of the intended scope of this study.*
>
> **Conclusions***: Other reasons might also be as important societal implication as improved navigability routes, but they are out of the scope of this study.*

RC1-2. The bifurcation function is adjusted from 70:30 in the present day to 50:50 in the historical figure, but the validity of this adjustment is unclear. This bifurcation function is crucial in determining the low flow of the divergent rivers. There is a risk that the stability of the low flow/navigation network of past divergent rivers is almost entirely determined by this function. When examining historical events, making such a bold assumption about this critical figure is questionable. At least some cases of this function need to be verified.

**Thank you for this comment. We agree with your point that this is an important assumption in our study. First, as we demonstrated in the text, the rate of 70:30 in the present day is reasonable from the viewpoint of river discharge time-series data analysis. As for river discharge 400 years ago, it is hard to know it because there was no quantitative observation left. One thing for sure is that the channel capacity of the Akahori River section, which connects the Edo River (the original southward route) and the Hitachi River (the new eastward route), was much smaller than today; hence most likely less than 70% of river water can travel the Akahori River.** We further justified the rationale of the rate in the revised text. Please note that a relevant sensitivity test has already been conducted, and we confirm that it does not influence our overall conclusions.

This text in the original manuscript (L258-264) was changed into the blue text below:

"Furthermore, we also conducted sensitivity analyses of both the T2A and T2E operational functions (data not shown) to determine the constant diversion rates (i.e., the percentages of river discharge flowing towards each river) when varying the percentages of river discharge flowing toward the Edo and Tone Rivers. The sensitivity analysis showed that the NSE values for Tone River observation stations downstream from Tonesekiyado were highest when the diversion rates were set to 70% of river discharge toward the Tone River and 30% toward the Edo River."

*Furthermore, we also analyzed both the T2A and T2E operational functions to determine the constant diversion rates (i.e., the percentages of river discharge flowing towards each river) by varying the percentages of river discharge flowing toward the Edo and Tone Rivers. The analysis showed that the NSE values for Tone River observation stations downstream from Tonesekiyado were highest when the diversion rates were set to 70% of river discharge toward the Tone River and 30% toward the Edo River.*

The blue part of the text below was added to the original manuscript (L367-370) in the revision:

"The rationale for choosing 70% flow toward the Tone River and 30% flow toward the Edo River at the T2E bifurcation was that the Kitasekiyado and Nishisekiyado stations should exhibit the best and near-equal performance after implementation of the bifurcation, which was achieved."

RC1-3. The historical river port locations are used as validation data, but the nature, validity, and reliability of this historical data need to be clarified. As the authors indicate, one of the critical contributions and challenges of this study is validating the simulation results in an era without modern river measurements. The reliability of this validation data could determine the significance of this study. Additionally, the process of developing historical data of river channels should be explained in more detail.

**Thank you for this suggestion. The nature, validity, and reliability of port location data are further disclosed in the revised text.**

**[Nature] The nature of the presented data is from the collective sources of historians, geographers, and other types of researchers who collected and reported those data. We carefully merged these resources and reconstructed our river port locations based on their reports.**

**[Validity] The validity of our presented river ports data arises from validations of the sources (i.e., ancient documents) who first reported them. Yet, some of them can be validated nowadays using geographical GIS and the name of "aza" mentioned below.**

**[Reliability] The reliability of our presented river ports data was tested by crosschecking several sources (i.e., books and papers) that reported similar data.**

**The following text was added in the manuscript to explain the nature, validity, and reliability of port location data:**

*The names of ports were taken from historians' books (e.g., Okuma, 1981) and confirmed by the name of "aza", an administrative designation of small sections into which some of the rural districts of Japan are divided. The old river routes were also taken from figures in books and digitalized to maintain the geometry of routes.*

**We agree that the process of developing historical data on river channels should be explained in more detail, below we provided clear explanations to precisely clarify the applied reconstruction processes of the historical maps.**

Currently, the process is explained in Chapter 3.3.3 Flow direction data:

L220-228: "Matsumura et al. (2021) developed a flow direction map for the present-day Tone, Edo, and Ara River basins with a resolution of 1 arc-min (approximately 2 km). During reconstruction of historical maps, the flow direction was manually changed at one or more points to match the available historical maps (i.e., MLIT, 2023c; Inazaki et al., 2014). When it was difficult to determine the exact historical river route (the available historical maps are often hand-drawn), both administrative borders and the associated terrain slopes were used as guidelines, because we assumed that these did not change over time even when the river routes varied. The parts of the domain that did not lie inside the three river basins were treated as no data points."

**Hereby, we added the following explanatory text to the revised manuscript to precisely define clarification of the reconstruction process of the historical maps, and we removed the underlined green text from the paragraph above:**

*The reconstructed digital historical maps are built from printed reconstructed flow direction maps in publications. As the base digital flow direction map, we used the one developed by Matsumura et al. (2021). Then, starting from the oldest reconstructed digital historical map in 1593, we have changed one or more flow directions on the way to complete TREDP by referring to available printed historical maps. During reconstruction of historical maps, the flow direction was manually changed at one or more points to match the available historical maps (i.e., MLIT, 2023c; Inazaki et al., 2014). When it was difficult to obtain the exact historical maps, then administrative borders were used as guidelines because we found that, when a river changes its route, its administrative borders usually do not change simultaneously with it. For instance, the flow direction for excavating Shinkawa-Dori diversion in 1621 was determined as the northern border between Saitama and Tochigi Prefecture (nowadays Watarase River), which remained the same until today, yet after the excavation of the Shinkawa-Dori the Tone River route moved slightly southward of the prefectural border. Another example is the diversion of Kinu River towards Kokai River in 1630, whose route followed the border between Moriya and Tsukubamirai Districts, which is a densely populated area nowadays. The parts of the domain that did not lie inside the three river basins were treated as no data points."*

**RC2:** ['Comment on egusphere-2024-595'](), Anonymous Referee #2, 13 Aug 2024

RC2-0. This manuscript presents an interesting study attempting to simulate the streamflow in a hydraulic engineering project influenced basin several centuries ago, and illustrate the values of Tone River Eastward Diversion Project in aiding navigation. The reconstructed streamflow is indirectly validated by some proxies, but no datasets of ancient hydrological and meteorological datasets are used to force or validate the model, which forms a significant disadvantage for a streamflow reconstruction study. However, I think this would be a small issue if the research topic can be adjusted slightly. Besides, I find some important methods/results are presented inadequately or unclearly. Consequently, I recommend a major revision before publication, by addressing the following major and specific comments.

> **Thank you for your useful evaluation and constructive suggestions. We hereby hope that you will find our response satisfactory. If there are some points that are still unclear or require further clarification, we will gladly address them with carefulness in the following round of revision.**
>
> **We replied to the referee comments, with our answer indicated by Answer (in green), corresponding actions indicated by Action (in blue), and textual changes in *italic font*:**

RC2-1. Expression of the research focus

When talking about historical streamflow reconstruction, we would certainly expect the reconstruction of the climate factors, since climate forcing is one of the most important factors influencing streamflow, leading to strong hydrological nonstationarity. This study mainly explores the streamflow produced by different historical maps. Although authors have acknowledged the limitation in this aspect, I still think the lack of climate reconstruction is an inevitable shortage. However, if we regard this study as an investigation on the influence of TREDP on streamflow, the lack of climate reconstruction would not be so important. Consequently, I suggest the authors to change the expression of the research topic, focusing on the influence of TREDP, rather than "reconstructing" the "historical" streamflow.

> **We appreciate your comment. We understand the word "reconstruct" infers a quantitative estimation of the past. We carefully revised the entire draft and made sure that the word was only used for river maps, with the exception of some cited studies, in which context the word was used for other purposes such as hydrological discharge or meteorological rainfall.**
>
> **We further clarified that streamflow reconstruction is not achieved in this study because of the lack of historical climate data, with the insertion of the following blue text, after the quoted green text from the original manuscript:**
>
> "Here and below, we note some important limitations of our study. First, the H08 historical maps were forced using present-day meteorological data. The flows may have differed from what we derived because the climatological conditions may have varied."

*Thus, although we claim that we accurately reconstructed historical river maps, this is not necessarily the case for streamflow due to different climatological factors from the past. Yet, we provided indirect validation by comparing the distribution of the Q20 river discharge with the most upstream navigable river ports in the past.*

**Your suggestion is actually already included in research question (3) of our study, which states: „What were the implications of TREDP? Were the implications consistent with the views of Koide and Okuma, who claimed that enhancement of navigation was key?"**

RC2-2. Model calibration

RC2-2-1. I am confused about the calibration procedure. The authors only adopt three different values for each procedure, totally 81 combinations. This seems not a normal calibration procedure, in which simulation discharge produced by a large number of parameters need to be calculated.

**We agree with you that the calibration procedure is different from many other conventional hydrological models. H08, including surface energy balance calculation, has long adopted this procedure and demonstrated it worked for many large basins (e.g., the Chao Phraya River in Thailand reported by Hanasaki et al., 2014; The Ganges Brahmaputra Meghna Rivers by Masood et al., 2015; The rivers in Kyushu island in Japan by Hanasaki et al., 2022). There are four most sensitive calibratable hydrological parameters and minimal three optional values (the minimum, mean, and maximum values of physical bounds for each parameter). Optional values can be increased (e.g., Masood et al. 2015 used five values for each). Since quite a limited number of parameters were tested, the parameter set found may not be the mathematical optimum, while the parameter set produces reasonable hydrological simulation in the entire basin (see text).**

RC2-2-2. Besides, the T2E bifurcation function is set as 70:30% because the NSE values are highest at this time (L262-264). Is NSE always highest at 70:30% for all the parameter combinations? Several lines afterward, the authors claim that the rationale for this value is the near-equal NSE of Kitasekiyado and Nishisekiyado stations. Is this consistent with the sensitivity analysis described previously?

**For your first question, our answer is no. NSE was highest at 70:30% only for the best hydrological parameter combination of H08 after the T2E bifurcation (see Figures 7b and 7d; the same NSE data are shown in Table 3 on the third and the fourth columns). For your second question, our answer is yes. We tested various combinations of the diversion ratio at the T2E bifurcation, and we selected the value of 70:30%, which exhibited near-equal NSE for Kitasekiyado (after T2E bifurcation, the eastern route) and Nishisekiyado (the southern route) stations for the best hydrological parameter combinations (NSE = 0.67 for both calibration and validation at Kitasekiyado and as NSE = 0.89 for calibration and 0.80 for validation at Nishisekiyado). The near-equal NSE at two stations implies a reasonable bifurcation over time.**

RC2-2-3. Shouldn't this bifurcation function be determined by something like project planning?

**Thank you for this suggestion. There must be a strict gate operation rule at the Sekiyado bifurcation point (so far, the authors have not yet accessed to the written rule yet), but it doesn't necessarily be the solution because the simulation includes bias and errors. Rather important is that the flow simulation at the upper and lower stations agrees well with the observation. We further revised the text to clarify our intention.**

This text in the original manuscript (L258-264) was changed into the blue text below:

"Furthermore, we also conducted sensitivity analyses of both the T2A and T2E operational functions (data not shown) to determine the constant diversion rates (i.e., the percentages of river discharge flowing towards each river) when varying the percentages of river discharge flowing toward the Edo and Tone Rivers. The sensitivity analysis showed that the NSE values for Tone River observation stations downstream from Tonesekiyado were highest when the diversion rates were set to 70% of river discharge toward the Tone River and 30% toward the Edo River."

*Furthermore, we also analyzed both the T2A and T2E operational functions to determine the constant diversion rates (i.e., the percentages of river discharge flowing towards each river) by varying the percentages of river discharge flowing toward the Edo and Tone Rivers. The analysis showed that the NSE values for Tone River observation stations downstream from Tonesekiyado were highest when the diversion rates were set to 70% of river discharge toward the Tone River and 30% toward the Edo River.*

The blue part of the text below was added to the original manuscript (L367-370) in the revision:

"The rationale for choosing 70% flow toward the Tone River and 30% flow toward the Edo River at the T2E bifurcation was that the Kitasekiyado and Nishisekiyado stations should exhibit the best and near-equal performance after implementation of the bifurcation, which was achieved."

RC2-3. Method description

The reconstruction of the historical maps seems to be a very important part in this study, because the simulation of historical streamflow is produced by simply replace the present map by historical ones. However, the method producing historical maps is only described by several simple sentences and referring to several literatures. I suggest authors to describe this part in details.

**We described the methodology for reconstructing the historical maps in section 3.3.3 (Flow direction data), but we agree that the explanation is insufficient. For instance, we stated that "administrative borders and the associated terrain slopes were used as guidelines because we assumed that these did not change over time even when the river routes varied", but we**

**did not provide concrete examples (i.e., Kinu and Kokai Rivers). In the revised manuscript, we enhanced explanations about methods used for reconstructing historical maps in more detail to the point of better methodological reproducibility.**

**We agree that the process of developing historical data on river channels should be explained in more detail, below we provided clear explanations to precisely clarify the applied reconstruction processes of the historical maps.**

Currently, the process is explained in Chapter 3.3.3 Flow direction data:

L220-228: "Matsumura et al. (2021) developed a flow direction map for the present-day Tone, Edo, and Ara River basins with a resolution of 1 arc-min (approximately 2 km). During reconstruction of historical maps, the flow direction was manually changed at one or more points to match the available historical maps (i.e., MLIT, 2023c; Inazaki et al., 2014). When it was difficult to determine the exact historical river route (the available historical maps are often hand-drawn), both administrative borders and the associated terrain slopes were used as guidelines, because we assumed that these did not change over time even when the river routes varied. The parts of the domain that did not lie inside the three river basins were treated as no data points."

**Hereby, we added the following explanatory text to the revised manuscript to precisely define clarification of the reconstruction process of the historical maps, and we removed the underlined green text from the paragraph above:**

*The reconstructed digital historical maps are built from printed reconstructed flow direction maps in publications. As the base digital flow direction map, we used the one developed by Matsumura et al. (2021). Then, starting from the oldest reconstructed digital historical map in 1593, we have changed one or more flow directions on the way to complete TREDP by referring to available printed historical maps. During reconstruction of historical maps, the flow direction was manually changed at one or more points to match the available historical maps (i.e., MLIT, 2023c; Inazaki et al., 2014). When it was difficult to obtain the exact historical maps, then administrative borders were used as guidelines because we found that, when a river changes its route, its administrative borders usually do not change simultaneously with it. For instance, the flow direction for excavating Shinkawa-Dori diversion in 1621 was determined as the northern border between Saitama and Tochigi Prefecture (nowadays Watarase River), which remained the same until today, yet after the excavation of the Shinkawa-Dori the Tone River route moved slightly southward of the prefectural border. Another example is the diversion of Kinu River towards Kokai River in 1630, whose route followed the border between Moriya and Tsukubamirai Districts, which is a densely populated area nowadays. The parts of the domain that did not lie inside the three river basins were treated as no data points."*

RC2-4. Figure 8 is an important figure showing the validation of historical simulation, but I find this figure and the interpretation on it extremely difficult to follow. What is the meaning and unit of the colorbar? What do the hollow circles represent? What do the channels with deep and shallow color in each map mean? The units of frame should be added. Also, I suggest the authors to refer to the specific point/line in the figure (e.g., … as shown by the XX point in Figure X) when describing this figure in the main text.

**We agree that Figure 8 was difficult to follow. Based on your suggestions, we believe that minor changes in the Figure is needed (e.g., adding LAT/LON units) but several insertions of new text in the Figure caption or in the main text of the manuscript is necessary for better understanding. Below are replies to all your questions.**

The main channel indicates river flow which meets the condition of Q20 > 20 $m^3$/s, which increases downstream. Q20 values ≤ 20 $m^3$/s are shown with yellow color.

The color bar indicates the Q20 values of river flow of the basin in the unit of $m^3$/s.

The hollow circles represent all recorded navigable historical ports except the most upstream navigable ones. They are only in Figure (8d) to not have too many circles overlapping.

Channels with pale colors represent river flow slightly greater than Q20 > 20 $m^3$/s, while the ones with dark blue colors indicate greater discharge up to maximal values shown in the color bar.

We added LAT/LON units to the frame.

**Below are our corrections made to the original Figure 8 caption.**

*The color bar indicates the Q20 values of river flow of the basin in the unit of $m^3$/s. The main channel indicates river flow which meets the condition of Q20 > 20 $m^3$/s, which increases downstream. Q20 values ≤ 20 $m^3$/s are shown with yellow color. Channels with pale colors represent river flow slightly greater than Q20 > 20 $m^3$/s, while the ones with dark blue colors indicate greater discharge up to maximal values shown in the color bar. The hollow circles represent all recorded navigable historical ports except the most upstream navigable ones.*

**We modified the manuscript and Figure 8 by properly specifying figures/panels/points wherever and whenever applicable.**

*Note that, in 1593, the eastern half of the present-day Tone River (the Hitachi River) and the western half of the Tone River were not connected and thus were unnavigable (Fig. 8a, the southeast part from the Nowata port). Moreover, the Q20 of the Hitachi River exceeded 20 $m^3$/s only after the confluence of the Kinu and Kokai Rivers, south from the Tatenuma port. In 1630, after diversion of the Kinu River, the section wherein Q20 exceeded 20 $m^3$/s moved upstream (Fig. 8b, south from the Tatenuma port). After the Sekiyado bifurcation (i.e., the old T2E) in 1666, the navigable section was substantially extended, now connecting the upper streams of major tributaries to the Pacific Ocean and Tokyo Bay (Fig. 8c, the part connecting southeast from the Nowata port and south from the Tatenuma port).*

RC2-5. -Figure 6: Maybe replace "before" and "after" by "without" and "with". "Before" and "after" seem to describe the time when an event happens, which may lead to misunderstanding.

Thank you. We agreed and corrected this for Figure 6 and Figure 7.

RC2-6. -L461: I think Figure 9 is just showing the simulated streamflow of different historical years, not a "validation".

Thank you. We agreed and corrected this.

RC2-7. -Figure 9: 1966 should be 1666?

Thank you. We agreed and corrected this.